# What Nutraceuticals Can Do for Duchenne Muscular Dystrophy: Lessons Learned from Amino Acid Supplementation in Mouse Models

**DOI:** 10.3390/biomedicines11072033

**Published:** 2023-07-19

**Authors:** Boel De Paepe

**Affiliations:** Department of Neurology, Ghent University & Neuromuscular Reference Center, Ghent University Hospital, Route 830, Corneel Heymanslaan 10, 9000 Ghent, Belgium; boel.depaepe@ugent.be

**Keywords:** N-acetylcysteine, amino acid supplements, L-arginine, Duchenne muscular dystrophy, mdx mouse, osmolytes, supportive therapy, taurine

## Abstract

Duchenne muscular dystrophy (DMD), the severest form of muscular dystrophy, is characterized by progressive muscle weakness with fatal outcomes most often before the fourth decade of life. Despite the recent addition of molecular treatments, DMD remains a disease without a cure, and the need persists for the development of supportive therapies aiming to help improve patients’ quality of life. This review focuses on the therapeutical potential of amino acid and derivative supplements, summarizing results obtained in preclinical studies in murine disease models. Several promising compounds have emerged, with L-arginine, N-acetylcysteine, and taurine featuring among the most intensively investigated. Their beneficial effects include reduced inflammatory, oxidative, fibrotic, and necrotic damage to skeletal muscle tissues. Improvement of muscle strength and endurance have been reported; however, mild side effects have also surfaced. More explorative, placebo-controlled and long-term clinical trials would need to be conducted in order to identify amino acid formulae that are safe and of true benefit to DMD patients.

## 1. Introduction

The severest form of muscular dystrophy (MD), termed Duchenne MD (DMD), is a progressive muscle-wasting disease. Its first symptoms usually occur at 2 to 3 years of age, and most patients become wheelchair-dependent when they reach their teens. In addition to accumulating skeletal muscle weakness, respiratory insufficiency and cardiomyopathy develop [1]. The disease is caused by mutations in the protein-coding *DMD* gene, which leads to the absence of functional dystrophin protein and the disconnection of the intracellular muscle fiber cytoskeleton from the extracellular matrix and the basal lamina. Via complex cascading pathways, sarcolemmal damage accumulates, resulting in muscle fiber necrosis and replacement with connective and fatty tissue [2]. The chronic inflammation that damages muscle tissues in DMD patients is counteracted by immunosuppressive therapy, which delays loss of ambulation and prolongs life expectancy, but comes with important side effects [3]. Despite intensive efforts in therapeutic innovation, DMD remains incurable to date. There are several promising novel therapeutic approaches for DMD that are entering the clinic; however, their limitations suggest that future disease management may still require the combination of molecular interventions with pharmacological agents [4]. The search for supportive therapies remains, therefore, valuable and worthwhile. Therapeutic development is facilitated by the availability of animal models such as the mdx mouse, the standard model for DMD. Despite genetic similarities to the human condition, the disease phenotype of mdx is much milder, exhibiting less severe skeletal muscle damage, with the notable exception of the diaphragm muscle [5].

A well-balanced diet is paramount to patient wellbeing, yet the progressive nature of DMD disease poses particular challenges. Many patients struggle with constipation and weight gain due to progressive muscle weakness and limited mobility. The prolonged therapeutic use of glucocorticoids further adds to the risk of developing obesity and related endocrine complications. Caloric intake needs to be balanced with physical activity to manage weight throughout the different disease stages, requiring personalized dietary advice and strategies adjusted throughout the patient’s life [6]. Recommendations to assess nutritional status and its associations with therapies, and guidelines for nutrient management and supplementation have been published [7]. Nutritional advice for DMD aims to preserve lean muscle mass and prevent excessive fat build-up and weight gain, assuring sufficient intake of protein. Other issues of concern are the prevention of osteoporosis and balanced calcium intake, and maintaining normal cholesterol and lipid levels through saturated fat moderation. Moderation of refined carbohydrate intake to avoid insulin resistance is recommended, and sufficient intake of dietary fibers is advised to prevent constipation. No international guidelines have been proposed for dietary supplements in DMD, except for the need for vitamin D and calcium supplementation when patients display a deficiency. The recommended daily allowance for vitamin D and calcium for DMD in boys from 4 years of age onward is 600 IU and 1 g, respectively [8]. Nonetheless, many patients use nutritional supplements without medical control. The potential benefits of nutraceuticals in DMD have been reviewed, providing a general overview of compounds to be considered [9], and their associated safety issues [10]. The purpose of this review is to evaluate the therapeutic potential of amino acid supplementation for DMD by analyzing available preclinical data obtained in the standard DMD disease model, i.e., the mdx mouse.

## 2. Amino Acid Supplementation as a Therapeutic Strategy for DMD

The human body needs amino acids to build up tissues, and for the regulation of a multitude of intracellular biochemical pathways. All natural amino acids concern the left-handed (L) stereoisomers. Their basic structure consists of four groups attached to a central carbon: a hydrogen, an α-carboxyl group, an α-amine group, and a side chain. Twenty two amino acids are proteinogenic, i.e., used to build up proteins. Instructions for the correct amino acid sequence that makes up a protein lie encoded in the DNA. tRNAs deliver amino acids to the ribosome, where polypeptides are generated by matching to the codons in mRNA. A subset of amino acids needs to be ingested via food, as they cannot be synthetized by human cells; these are termed essential amino acids. The essential amino acids are histidine (His), isoleucine (Ile), leucine (Leu), lysine (Lys), methionine (Met), phenylalanine (Phe), threonine (Thr), tryptophan (Trp), and valine (Val). The remaining nonessential proteogenic amino acids are glutamine (Gln), aspartate (Asp), glutamate (Glu), arginine (Arg), proline (Pro), cysteine (Cys), selenoCys, asparagine (Asp), serine (Ser), glycine (Gly), and tyrosine (Tyr). Amino acids can become chemically modified after they have been incorporated into a protein through posttranslational modification, most importantly through phosphorylation, carboxylation, and hydroxylation, strongly influencing protein function, localization, and cell signaling. A subset of amino acids are termed conditionally essential, including the 2-aminoethanesulfonic acid termed taurine (Tau) and Cys [11]. Semi-essential amino acids need to be ingested to uphold sufficient levels at set times, during infancy, or in certain health conditions. Diverse naturally occurring amino acid-like structures exist, and many more can be synthetically engineered.

Balanced amino acid levels are a prerequisite for the preservation of protein homeostasis, hence the correction of the amino acid level disturbances associated with DMD is an amenable therapeutic strategy. In this respect, decreased circulating Tau levels have been recorded in wheelchair-bound but not in younger ambulant DMD patients, when compared to age-matched healthy boys [12]. Altered amino acid contents have also been observed in a mouse model, displaying age- and muscle group-dependent differences and differences between studies. In quadriceps of 3 months and 6 months old mdx, Glu and Gln levels were higher, and Gly levels were lower than in controls. In the diaphragm, higher levels of Glu, Gln, and Ile were observed in three-month-old mdx and Ala, Glu, and Ile in six-month-old mdx. Met and Tau levels in diaphragm muscles were lower in the six-month-old mdx compared to control animals [13]. In quadriceps of seven-week-old mdx, the levels of Met, Gly, Glu, and Tau were reduced; however, by the age of seventeen weeks Gly and Glu levels became higher than in control mice. These authors reported Met and Tau levels were lower in mdx quadriceps of both ages [14]. Another study, however, found that Tau levels were low in mdx of 4 weeks old, and not at the age of 6 weeks [15]. From the above studies, complex dysregulation of amino acid levels in different muscle groups and at different ages became apparent.

Many amino acids display synergistic antioxidant activity, i.e., they enhance the effects of primary antioxidants via chelation of pro-oxidative metal traces and via regeneration of oxidized primary antioxidants. Within the proteogenic amino acids, Trp, Met, His, Lys, Cys, Arg, and Tyr have greater antioxidative capacities. The natural non-proteinogenic amino acid β-Alanine has been proposed as an especially potent scavenger of reactive oxygen species (ROS), with positive effects on muscle performance [16]. The non-essential amino acid Cys is required in the synthesis of the major intracellular antioxidant glutathione [17]. The Cys derivative N-acetylcysteine (NAC) has antioxidant properties, both directly, exerted through ROS scavenging by its thiol group, and indirectly, as a precursor of Cys. NAC has been observed to reduce oxidative damage and inflammatory changes in muscle cells in vitro [18], and to improve fatigue resistance and increase Cys levels in skeletal muscle tissues [19].

Various amino acid-like structures possess osmoprotective activities and are termed osmolytes. Tau is the most studied osmolyte in the context of DMD. Promising results continue to accumulate for Tau treatment in mdx mouse models, showing improvement of muscle strength and reduction of tissue inflammation and oxidative stress [20]. The osmoprotective Gly derivative trimethylglycine (TMG), or betaine, has also been observed to improve muscle performance [21]. TMG is obtained from the diet or via choline oxidation, and becomes further oxidized in the liver, thus forming the osmolytes dimethylglycine and trimethylamine-N-oxide [22]. The mechanism by which TMG supplementation affects muscle strength is elusive, but is presumed to be related to an increase in creatine, the energy substrate for muscle contraction [23]. Interestingly, the ratio of TMA to creatine correlates with muscle function and is decreased in DMD patients [24]. The osmolyte 1,4,5,6-tetrahydro-2-methyl-4-pyrimidinecarboxylic acid or ectoine (Ect) is a cyclic amino acid derived from extremophiles with therapeutic potential. Ect is a potent cell protectant that can be accumulated or removed to counteract osmotic stress. It forms a hydration shield that protects the functionality of macromolecules inside cells, acting against protein denaturation and misfolding [25].

Often, the osmoprotective and antioxidant activities of amino acids are observed to join forces. Tau, for instance, also possesses potent thiol antioxidant activity in muscle tissue [26]. Cys and its derivative L-2-oxothiazolidine-4-carboxylate (OTC) are metabolized to Tau, and OTC treatment increases Tau content, thereby improving muscle function in mdx [27]. As high doses become cytotoxic, tissue Cys content is under tight regulation, with excess Cys converted to Tau [28]. Tau also protects against the loss of muscle mass and reduced functionality associated with aging [29]. The protective effects of Tau have now been shown to also include preserved calcium homeostasis, membrane stabilization, and attenuation of apoptosis. Further, as Tau is able to restrain nuclear factor κB (NFκB) activity, a central regulator of tissue inflammation [30], Tau supplementation has been proposed as a more general anti-inflammatory strategy for various chronic inflammatory diseases.

The described complex multifaceted cell protective effects of amino acids can be of benefit to ameliorate muscle tissue stress in human diseases, and could therefore be considered a supportive therapy for DMD.

## 3. Pre-Clinical Studies Evaluating Amino Acids and Derivatives in Mdx

The most widely used animal model for studying DMD is the mdx mouse. Mdx are of C57BL/10ScSn background and carry a nonsense point mutation (C-to-T transition) in exon 23 of the *dmd* gene [31]. Even though mdx mice are equally as dystrophin protein-deficient as DMD patients, the disease phenotype is much milder. Unlike the human disease course, skeletal muscle from 12 weeks onwards achieves relative stability of function, owing to robust tissue regeneration. Only the diaphragm shows progressive deterioration, mimicking progressive disease in affected humans. Other animal models are available, including mdx mice of a genetic DBA2/J background (D2-mdx), displaying more severe muscle function impairment and tissue fibrotic damage [32]. The cardiac phenotype of mdx is also less severe compared to human disease, which often progresses toward a lethal dilated cardiomyopathy [33]. Cardiac compensatory processes appear to depend on preserved mitochondrial content and increased efficiency of oxidative phosphorylation, driving successful regulation of calcium homeostasis in heart cells in the early mdx disease stages [34]. Nonetheless, mdx mice remain a valuable model to test therapeutic interventions, not in the least because of our extensive knowledge of the model. The mdx disease phenotype can be aggravated by subjecting mice to strenuous exercise. Other strategies to achieve disease characteristics more resembling severe disease in humans are the genetic elimination of utrophin and α7-integrin, generating double-knockout strains with aggravated muscle disease; however, such models are difficult to breed and maintain.

Available published results on the effects of amino acid supplementation in mdx describing the effects on skeletal muscle function and tissue characteristics are summarized herein (see Table 1) [27,35,36,37,38,39,40,41,42,43,44,45,46,47,48,49,50,51,52,53,54,55,56,57].

Preclinical studies in mdx have identified compound-specific beneficial effects on skeletal muscle tissue properties (Figure 1). In addition, attenuated myocardial inflammation and fibrosis and improved cardiomyocyte calcium handling have been described after NAC treatment [58]. NAC has also been observed to improve lung function in mdx [51]. However, in addition to individual effects, combinations of amino acids may have added value as a therapeutic strategy. In this respect, the essential amino acids Val, Leu, and Ile, jointly termed branched-chain amino acids (BCAA), are often combined. Experimental evidences suggest that in various human diseases, BCAA levels are reduced [59]. Reduced levels of BCCA may affect the body glutamate–glutamine pool, leaving the tissue more vulnerable to oxidative stress, and in such incidences, BCAA supplementation may be beneficial. The anabolic properties of BCAA appear especially useful for ameliorating skeletal muscle atrophy via a dual mechanism: stimulation of protein synthesis on the one hand, through activation of the mTOR signaling pathway, and inhibition of protein breakdown on the other hand, through curbed Atrogin-1 and MuRF-1 expression reducing Ub-proteasome activity [60]. Through stimulation of protein synthesis, BCAAs have been observed to ameliorate exercise performance, enhance muscle strength, and reduce fatigability [61,62], with positive effects on stamina also observed in mdx [57].

## 4. Combining Amino Acid Supplements with DMD Standard of Care

Nutraceuticals can never be curative, and can only offer valuable support to patients’ overall wellbeing. It is therefore paramount to evaluate how they perform in combination with the standard of care for DMD, as well as with the treatment regimens of the future. With optimal management of cardiopulmonary dysfunction, patients with DMD can nowadays survive to live in their forties. Immunosuppression represents a cornerstone of therapy for achieving increased life expectancy, yet the spectrum of adverse reactions associated with long term steroid use is an incentive for the development of alternative compounds [63]. Meanwhile, nutritional supplements could counteract some glucocorticoid-induced side effects, which include osteoporosis, obesity, short stature, delayed puberty, and adrenal insufficiency.

Evidencing their positive effects on weight and metabolism, mice fed with a diet enriched with free essential amino acids were leaner and had improved metabolic parameters [64]. An inverse association between higher dietary BCAA intake and odds of obesity has also been described in humans [65]. Nonetheless, it is heavily debated whether BCAA protect against or promote insulin resistance. Either way, a large number of studies confirm an association between increased plasma levels and insulin resistance, appointing BCAA a role as biomarkers [66]. The impact of BCAA supplementation on metabolism is complex, and remains partly elusive. In mice on a high-fat diet, adding BCAA to the chow improves insulin tolerance, yet this effect is completely reversed by exercise training [67]. In obese humans on a hypocaloric diet, BCAA supplementation was not observed to affect lean muscle mass or insulin sensitivity, and a high-protein diet was suggested to be more advantageous [68].

Several amino acids and amino acid metabolites are associated with bone health and could be considered as dietary supplements to prevent osteoporosis. Supplementing essential amino acids increased bone strength in rats on an isocaloric diet [69]. A study reported BCAA and Ala in women and Trp in men as the most important amino acids inversely associated with osteoporosis in people of advanced age [70], and higher Trp serum levels predicted a low risk of osteoporotic fractures [71]. Oxidized Trp promotes bone marrow stem cells’ differentiation into osteoblasts [72].

Another possible bonus of amino acids could be the enhancement of anti-inflammatory effects. One could speculate that supplementation would reduce the necessary doses of glucocorticoids and/or enhance the therapeutic efficacy. The feasibility of this strategy is supported by two studies in mdx reporting that combining deflazacort with an Arg supplement improves histological and functional characteristics [39], and Gly supplementation augments the benefits of prednisolone treatment to diaphragm fibrosis [45].

Molecular therapies with different mechanisms of action are currently entering the clinic for DMD at an accelerated pace. In a first molecular strategy, the splicing process of dystrophin is altered, bypassing an out-of-frame mutation with antisense oligonucleotides [73]. Preservation of the dystrophin reading frame allows for the production of partially functional dystrophin, thereby reducing the severity of the disease phenotype. Several such exon-skipping therapies have been approved by the Food and Drug Administration to date: eteplirsen [74], golodirsen [75], viltolarsen [76], and casimersen [77], generating slight improvements in life expectancy. The interactions of exon-skipping therapies with dietary supplements are largely unknown. One study observed co-administering Gly in mdx led to increased uptake of phosphorodiamidate morpholino oligomers in regenerating muscle fibers mediated through mTORC1 activation, resulting in functional improvement and a spectacular 50-fold increase in dystrophin expression in abdominal muscles [78]. In addition, intravenous supplementation with Gly 7 days prior to exogeneous muscle stem cell or primary myoblast transfer in mdx augmented the efficiency of transplantation, which was observed in the increased numbers of dystrophin-positive muscle fibers evaluated in the tibialis anterior after 3 weeks. A protocol for Gly-enhanced exon-skipping has recently been published [79]. Lower dosages of exon-skipping therapies could substantially reduce their cost and their risk of side effects. Another promising molecular approach is the transfer of a microdystrophin gene copy inserted in an adeno-associated virus (AAV) vector [80]. On 22 June this year, the U.S. Food and Drug Administration approved elevidys, the first gene therapy for the treatment of pediatric patients of 4 to 5 years of age. Gene replacement therapy for DMD is currently still mostly in the experimental phase, and the effects of amino acid supplementation on therapeutic efficiency have not been evaluated.

## 5. Adverse Effects and Human Clinical Trials

Amino acid supplements are commonly used by healthy individuals as well as patients. However, enhanced intake of amino acids is by no means risk-free, and may entail side effects, especially at high doses or after prolonged use [81]. Intravenous administration of amino acid supplements may cause more significant adverse effects when compared to oral or inhaled administration. For DMD, further research is necessary to determine which amino acid supplements are beneficial, and what their interactions with therapeutic interventions are.

From studies in mdx mouse models, mild adverse effects have surfaced (Table 1). The most notable side effects of reduction in weight and developmental delay appear to be dependent upon dose, route of administration, and study-specific conditions. For example, Tau in high doses was described to cause growth deficits [55] while lower doses did not [52], yet another study did not show deleterious effects with the same high dose of Tau [53]. Impaired body weight gain was also reported in NAC-treated young growing mdx [49]. NAC has been used for years for different medical indications, most particularly in respiratory medicine for treating chronic obstructive pulmonary disease, interstitial lung diseases, bronchiectasis, and infectious disease. Additionally, NAC is used as an antidote for paracetamol poisoning, and in psychiatric illnesses and addictive behaviors [82]. It is considered a “conditionally essential” amino acid by some because the synthesis of Cys may be compromised under the stress of illness or in preterm infants [11]. For chronic use, a maximum dose of 600 mg/day is maintained, with the main indication being chronic obstructive pulmonary disease. Most of the side effects related to the oral intake of NAC are associated with gastrointestinal symptoms and nausea. Interestingly, Arg supplements have been observed to alleviate gastrointestinal dysfunction in mdx [83].

In comparison to relatively abundant studies in the mdx model, few human trials have been attempted with amino acid supplements in DMD patients [10]. From the available evidence, we can conclude that some benefit of amino acid supplementation may be a general phenomenon, irrespective of specific formulae. For instance, in a randomized, double-blind study comparing an oral supplement of Gln (0.5 g/kg/day) to a nonspecific amino acid mixture (0.8 g/kg/day), the latter equally inhibited whole-body protein degradation in DMD [84]. However, a double-blind, randomized crossover trial with sequential intervention periods of 0.5 g/kg/day Gln and placebo for 4 months did not observe any clinical benefit [85]. Some other clinical trials have also generated disappointing results. A randomized, placebo-controlled study evaluating supplementation with the amino acid derivate creatine 5 g/day for 8 weeks and determining calf muscle phosphorus metabolite ratios, provided no evidence for the benefit of long-term treatment, nor did it observe an effect on DMD patient lifespan [86]. A double-blind, placebo-controlled clinical trial evaluating Gln (0.6 g/kg/day) and creatine (5 g/day) for 6 months found no statistically significant effect on muscle strength [87]. Another randomized clinical trial reported more promising results in ambulant DMD patients aged 6.5 to 10 years treated with a combination of 2.5 g L-citrulline and 0.25 g metformin three times a day for 26 weeks. A clinically relevant reduction in motor function decline was observed, with favorable effects reaching significance only in stable patients. Overall, treatment was well tolerated, with only mild adverse effects reported [88].

NAC has not been tested in DMD; however, a randomized double-blinded placebo-controlled trial was conducted in another genetic muscle disorder, i.e., ryanodine receptor 1-related myopathy. Oral NAC daily for 6 months (adults 2.7 g; children 30 mg/kg) did not lead to adverse events; however, therapeutic benefit was equally lacking, with no improvement in participants’ 6 min walk test distance and unchanged levels of oxidative stress [89].

No results of clinical trials of Tau supplementation have been reported for DMD either; however, studies in metabolic disorders have generated encouraging results. In type 2 diabetes mellitus patients on a low-calorie diet, placebo-controlled administration of 1 g Tau three times per day for 8 weeks lead to a considerable decrease in serum insulin, along with improvement in inflammatory and oxidative stress markers [90]. In an open-label, phase III trial in mitochondrial myopathy, 9 to 12 g Tau per day for 52 weeks effectively reduced the recurrence of stroke-like episodes in patients suffering from lactic acidosis and stroke-like episodes (MELAS) [91]. Additionally, clinical trials have already picked up on the positive effects of Tau on muscle function. For instance, in a double-blinded randomized clinical trial, patients with chronic liver disease receiving 2 g Tau per day as an oral supplement self-reported a significant reduction in the frequency, duration, and intensity of muscle cramps [92].

Tau supplementation may transcend therapeutic use in human disease. In healthy subjects, benefits of Tau supplementation on aerobic and anaerobic performance, muscle damage, metabolic stress, and recovery have been reviewed recently, concluding that evidence so far shows varied and mostly limited effects [93]. However, a placebo-controlled evaluation of single oral administration of 0.1 g/kg Tau before resistance training did observe a significant positive effect on exercise performance [94], and a meta-analysis of ten published studies concluded that Tau supplements of 1 to 6 g/day for 2 weeks did improve endurance performance [95]. Overall, Tau supplementation was well tolerated, as no severe adverse events have been reported. Of interest, Tau abundance decreases during aging, and a reversal of this decline through Tau supplementation has very recently been reported to increase health and/or life span in different species including monkeys [96].

## 6. Conclusions

The desired properties of supportive therapies for DMD include anti-inflammatory [97], anti-oxidant [98], and anti-fibrotic [99] activities. Amino acids and derivatives have been shown to possess combinations of these characteristics, making them plausible candidates for disease-modifying nutraceuticals. Preclinical studies in mdx mouse disease models have increased our understanding of their benefits and associated side effects, and have identified promising candidates. Arg, NAC, and Tau are among the most intensively investigated, and their compatibility with the current standard of care, glucocorticoid and exon-skipping therapies, has been investigated to some extent. Only a limited set of supplements have, however, been tried in humans up to now; therefore, not enough evidence is currently available to propose amino acid formulae that are of true benefit to DMD patients. More explorative, placebo-controlled and long-term clinical trials are required.

## Figures and Tables

**Figure 1 biomedicines-11-02033-f001:**
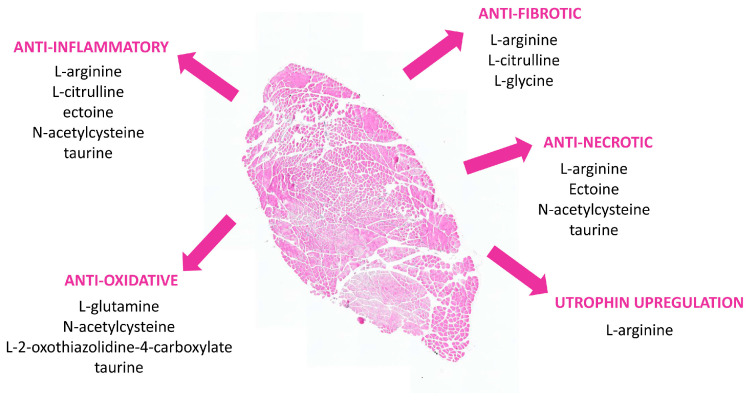
Beneficial effects of amino acids and derivatives on mdx skeletal muscle tissue histopathology, as observed in preclinical studies. Illustrative image shows a hematoxylin and eosin-stained cross-section of a full tibialis anterior muscle from a 26-week-old mdx mouse.

**Table 1 biomedicines-11-02033-t001:** Comprehensive summary of functional and histological skeletal muscle tissue responses to amino acid supplementation in the Duchenne mouse model.

Supplement	Mice	Administration and Dose	Muscle Outcome	Reference
β-Alanine	Male mdx untreated (n = 8) β-Ala-treated (n = 8); age 20 weeks.	3% in drinking water for 4 weeks.	Increased resistance to fatigue after intermittent electrical stimulation for 1 min in EDL ex vivo.	[35]
L-Arginine	Male mdx saline-treated (n = 7) Arg-treated (n = 7); age 8 weeks.	Intraperitoneal injection of 0.4 g/kg/day for 4 weeks.	Improved contractile properties of EDL. Increased utrophin and ϒ-sarcoglycan protein levels in EDL.	[36]
	Female mdx saline-treated (n = 8) Arg-treated (n = 8); age 16 weeks.	Intraperitoneal injection of 0.2 g/kg/5 days a week for 6 weeks.	Higher isometric twitch tension in DIA ex vivo. Reduced necrotic surface in TA, GAS, EDL, DIA. 3-fold increased utrophin protein levels in SOL, EDL, DIA, TA.	[37]
	Male mdx untreated (n = 7) Arg-treated (n = 7); age 5 weeks.	Intraperitoneal injection of 0.20 g/kg/day for 2 weeks.	Decreased non-muscle areas and enhanced muscle regeneration in DIA. Decreased levels of TNFα, IL-1β, IL-6, NFκB protein in DIA.	[38]
	Mdx untreated (n = 8) Arg-treated (n = 8); age 4 weeks.	0.375% in drinking water + 1.2 mg/kg/day deflazacort for 3 weeks.	Increased distance running capacity immediately after and 3 months after treatment. Attenuated exercise-induced damage and regeneration of QUA.	[39]
	Mdx saline-treated (n = 6) Arg-treated (n = 6); age 1 week.	Intraperitoneal injection of 0.8 g/kg/day for 6 weeks.	Reduced magnitude of contraction-induced force drop in TA in vivo. Lower level of central nucleation of muscle fibers in TA.	[40]
L-Carnitine	Mdx untreated (n = 5) Carnitine-treated (n = 5) age 3 weeks.	Oral 0.75 g/kg/day for 6 weeks.	Higher exercise tolerance and lower blood CK after 30 min horizontal treadmill exercise. Less severe QUA sarcolemmal disruption after 30 min strenuous eccentric exercise.	[41]
L-Citrulline	MDX age 4–5 weeks.	2 g/kg/day for 8 weeks.	Improved increment of maximal forelimb strength. Increased specific isometric twitch and tetanic force in DIA ex vivo. Reduced inflammation and fibrosis in GAS and DIA.	[42]
Ectoine	Male and female mdx untreated (n = 10) Ect-treated (n = 11); age 1 week.	0.5% (1.1 g/kg/day) in drinking water for 5 weeks.	Decreased CCL2, TNFα, and IL1β expression in TA. Increased amount of healthy fibers in TA.	[43]
	Male and female mdx saline-treated (n = 8) Ect-treated (n = 9); age 1 week.	0.075% in drinking water (0.2 g/kg/day) for 2 weeks, followed by intraperitoneal 0.2 g/kg/day for 2 weeks.	Increased amount of healthy fibers in TA.	[43]
L-Glutamine	Mdx saline-treated (n = 4) Gln-treated (n = 4); age 4 weeks.	Intraperitoneal injections of 0.5 g/kg/day for 3 days.	Decreased ERK1/2 activation and oxidative stress in QUAD	[44]
L-Glycine	Male mdx and mdx:utrophin−/− controls treated with 1.9% Ala in drinking water (2 × n = 8) Gly-treated (2 × n = 8); age 4 weeks.	1.9% (2.5 g/kg/day) for 8 weeks (mdx) or 14 weeks (mdx:utrophin−/−).	Reduced fibrosis in DIA.	[45]
NAC	Male mdx untreated (n = 9) NAC-treated (n = 6); age 3 to 9 weeks.	1% in drinking water for 6 weeks; Ex vivo perfusion of EDL with 20 mM NAC.	Fewer centrally located myonuclei, reduced ROS, decreased nuclear NFκB, and increased utrophin expression in EDL. Greater force value of EDL ex vivo.	[46]
	Male mdx untreated (n = 11) NAC-treated (n = 8); age 6 weeks.	1% in drinking water for 6 weeks.	Prevention of exercise-induced (30 min horizontal treadmill) muscle fiber necrosis in QUA.	[47]
	Male mdx untreated (n = 11) NAC-treated (n = 11), age 11 weeks.	4% (2 g/kg/day) in drinking water for 1 week.	Reduced CK increase after exercise. Prevention of exercise-induced (30 min horizontal treadmill) muscle fiber necrosis in QUA. Decreased protein thiol oxidation in QUA.	[48]
	Male mdx saline treated (n = 10) NAC-treated (n = 10); age 2 weeks.	Intraperitoneal injection of 0.15 g/kg/day for 2 weeks.	Reduced blood CK. Decreased sarcolemmal leakage and muscle fiber necrosis in DIA. Reduced TNFα levels in DIA.	[49]
	Male mdx untreated (n = 8) NAC-treated (n = 8); age 6 weeks.	2% in drinking water for 6 weeks.	Lower body weight, lower EDL muscle weight. Greater normalized forelimb grip strength. Unchanged ex vivo EDL muscle force. Activity of macrophages decreased in GAS muscle. Reduced protein thiol oxidation in EDL.	[50]
	Male mdx untreated (n = 10) NAC-treated (n = 10); age 8 weeks.	1% in drinking water for 2 weeks.	Improved force-generating capacity. Reduced immune cell infiltration and collagen deposition; reduced IL-1β and CXCL1 levels in DIA.	[51]
	Male mdx untreated (n = 6) and NAC-treated (n = 6); age 3 weeks.	2% in drinking water for 6 weeks.	Blunted growth and reduced EDL muscle weight. Unchanged maximum specific force of EDL ex vivo. Reduced abnormal fiber branching and splitting in EDL.	[27]
OTC	Mdx untreated (n = 6–8) OTC-treated (n = 6–8); age 6 to 12 weeks.	0.5% in drinking water for 6 weeks.	Increased forelimb grip strength. Reduced protein oxidation in QUA.	[27]
	Male and female mdx untreated (n = 6) and OTC-treated (n = 8); age 2.5 weeks.	0.8 g/kg/day for 3.5 weeks.	Improved normalized forelimb grip strength. Increased maximum specific force of EDL muscle ex vivo. Decreased CSA of EDL.	[52]
Taurine	Male mdx untreated (n = 5) Tau-treated (n = 5); age 3–4 weeks.	Male mdx untreated (n = 5) and Tau-treated (n = 5); age 3–4 weeks. Subjected to chronic exercise on a treadmill.	Ameliorated negative threshold voltage values of EDL fibers.	[53]
	Male mdx untreated (n = 8) Tau-treated (n = 8); age 20 weeks.	3% Tau in drinking water for 4 weeks.	Decreased body mass and EDL muscle mass. Increased recovery force production and increased resistance to fatigue after intermittent electrical stimulation for 1 min in EDL ex vivo.	[35]
	Male and female mdx untreated (n = 6) and Tau-treated (n = 8); age 2.5 weeks.	4 g/kg/day for 3.5 weeks.	Decreased CSA of EDL. Three-fold decreased protein thiol oxidation in EDL.	[27]
	Male mdx vehicle (n = 19) and Tau-treated (n = 9); age 4–5 weeks.	1 g/kg/5 day a week in drinking water for 4 weeks.	Improved muscle force after exercise. Reduced percentages of damaged area and NFκB-positive myonuclei in GAS. Reduced ROS production in TA.	[54]
	Male and female mdx untreated (n = 10) and Tau-treated (n = 8); age 1 week.	8% (16 g/kg/day) in drinking water for 5 weeks.	A 12% reduced tibia length and 25% reduced CSA of EDL. Some 20% reduced protein thiol oxidation in EDL.	[55]
	Male mdx untreated (n = 14) and Tau-treated (n = 10) prior to conception.	2.5% in drinking water evaluated at 4 and 10 weeks of age.	A 50% reduction in non-contractile tissue in TA muscle at 4 weeks, but no change at 10 weeks.	[56]
	Male and female mdx untreated (n = 10) and Tau-treated (n = 11); age 1 week.	2.5% (4.6 g/kg/day) in drinking water for 5 weeks.	Decreased CCL2 and SPP1 expression in TA.	[43]
Branched-chain	Male and female mdx untreated (n = 10) and BCAA-treated (n = 10) mdx; age 12 weeks.	1.5 g/kg/day in drinking water for 2 weeks.	A 20% increased endurance time on the treadmill. Higher numbers of slow fibers in TA and VM.	[57]

Abbreviations: β-Alanine (β-Ala), L-arginine (Arg), branched-chain amino acids (BCAA),, Ectoine (Ect), L-glutamine (Gln), L-glycine (Gly), N-acetylcysteine (NAC), L-2-oxothiazolidine-4-carboxylate (OTC), taurine (Tau); DIA diaphragm (DIA), extensor digitorum longus (EDL), tibialis anterior (TA), quadriceps (QUA), vastus medialis (VM); cross-sectional area (CSA). Mdx are C57Bl/10ScSnmdx/mdx, unless otherwise specified. Branched-chain amino acids are L-valine, L-leucine, and L-isoleucine.

## Data Availability

No new data were created or analyzed in this study. Data sharing is not applicable to this article.

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
