# Peer review of "What Nutraceuticals Can Do for Duchenne Muscular Dystrophy: Lessons Learned from Amino Acid Supplementation in Mouse Models"

_biomedicines, 2023, doi:10.3390/biomedicines11072033_

Round 1
Reviewer 1 Report
This is a qualitative review written by a specialist in the field of muscular dystrophies. I have a couple of recommendations.
Page 4. Regarding mdx mice, I would recommend the author to separately focus on the problem of the development of cardiac pathology. It develops very slowly in mdx mice, in particular, due to compensatory processes (this also applies to ion transport in cardiomyocytes and higher mitochondrial activity (especially in young mice), which delays the development of cardiomyopathy).
Also in this sense, I would like to recommend the author to separately highlight the effect of amino acids on cardiac function, if possible. In this version, the author has focused only on muscle function in general.
It is also necessary to provide at least one general figure showing the various uses of amino acids for the treatment of DMD. This will greatly increase the attractiveness of the work. This also applies to graphical abstract.
Author Response
Dear reviewer,
Many thanks for your constructive remarks, which allowed improvement of the manuscript. I have incorporated the recommendations to the best of my abilities, and hereunder reply to the remarks point-by-point.
1) Page 4. Regarding mdx mice, I would recommend the author to separately focus on the problem of the development of cardiac pathology. It develops very slowly in mdx mice, in particular, due to compensatory processes (this also applies to ion transport in cardiomyocytes and higher mitochondrial activity (especially in young mice), which delays the development of cardiomyopathy).
Apologies for not sufficiently recognizing cardiomypathy as an important aspect of DMD pathogenesis. Indeed, as a lead cause of death, heart failure is an important drug target. Unfortunately, my own experience and expertise do not allow me to go into detail on cardiac function in mdx and preclinical studies of beneficial interventions. Perhaps this would be a subject on its own, in a review article to be construed by a leader in the field. To accommodate the remark, the revision contains description of the mdx model cardiac phentype, and new references 31 and 32 have been added in evidence.
2) Also in this sense, I would like to recommend the author to separately highlight the effect of amino acids on cardiac function, if possible. In this version, the author has focused only on muscle function in general.
The title of Table 1 has been changed to reflect the focus which is placed on skeletal muscle specifically. The text of Section 3 now includes a study of NAC-mediated improvement of mdx cardiac phenotype, i.e. new reference 57.
3) It is also necessary to provide at least one general figure showing the various uses of amino acids for the treatment of DMD. This will greatly increase the attractiveness of the work. This also applies to graphical abstract.
Thank you so much for this suggestion; I completely agree that including a figure complements the reading experience considerably. I have included Figure 1 in the revised manuscript, which illustrates the different aspects of benefit of amino acids on skeletal muscle histopathology as observed in mouse studies.
Reviewer 2 Report
This review is quite captivating and exhibits a considerable level of scholarly writing. I would like to bring attention to several minor points that warrant consideration:
1. On page 6, providing an explanation for the abbreviations Smax and Vmax would greatly enhance clarity and comprehension.
2. In section five, I strongly recommend the inclusion of Table 2, mirroring the structure of Table 1, to offer a concise summary of findings from human participant-based studies. This table should effectively outline the potential benefits and adverse effects of amino acid supplementation in these studies. Additionally, it would be advantageous to ensure Table 1 is succinct and streamlined.
3. Emphasizing reference 76 in close proximity to lines 85-86 would serve to underscore the conditional essential nature of Cysteine, bolstering the significance of this point.
4. Near line 62, I propose the incorporation of a succinct summary outlining the guidelines or suggestions for vitamin D and calcium supplementation. It is advisable to support this information with an appropriate reference.
Author Response
Dear reviewer,
Thank you for your comments, which I have incorporated in de revised manuscript and address hereunder point-by-point.
1) On page 6, providing an explanation for the abbreviations Smax and Vmax would greatly enhance clarity and comprehension.
Based upon the remarks made by reviewer 1, I have renamed the table to reflect focus on skeletal muscle specifically. In this respect, description of respiratory benefit has been omitted from the table, and has been moved to the text, mentioned along with effects on cardiac function.
2) In section five, I strongly recommend the inclusion of Table 2, mirroring the structure of Table 1, to offer a concise summary of findings from human participant-based studies. This table should effectively outline the potential benefits and adverse effects of amino acid supplementation in these studies. Additionally, it would be advantageous to ensure Table 1 is succinct and streamlined.
Thank you for your suggestion. Indeed, I have considered including a comparative table with clinical trial data, however two main arguments made me decide against. Firstly, a review published in 2020 includes a table elegantly listing also clinical trials in DMD testing nutritional supplements, reference 10 in the manuscript. As there is little to add at this point, I would prefer to cite the review for readers to consult. Secondly, with this review I wanted to focus in-depth on preclinical studies as conveyed in the title, of course without ignoring clinical trials. The manuscript briefly described clinical trial results in section 5. If you insist on including an extra table, I would be willing to oblige.
3) Emphasizing reference 76 in close proximity to lines 85-86 would serve to underscore the conditional essential nature of Cysteine, bolstering the significance of this point.
As requested, I have cited the conditional essential behaviour of cysteine earlier in the manuscript in the revised version, moving the reference from 76 to 11.
4) Near line 62, I propose the incorporation of a succinct summary outlining the guidelines or suggestions for vitamin D and calcium supplementation. It is advisable to support this information with an appropriate reference.
Guidelines for vitamin D and calcium RDA have been added and new reference 8 has been introduced in evidence.
Reviewer 3 Report
The manuscript is interesting, I have some suggestions:
1) Abstract. L16-19. Beneficial effects include improved muscle function and reduced muscle fiber necrosis and reduced inflammatory and oxidative damage to muscle tissues, however mild side effects have also surfaced. In order to identify amino acid formulae that are safe and of true benefit to DMD patients, more explorative, placebo-controlled and long-term clinical trials need to be conducted, documenting therapeutic outcomes. Could you please improve the description of conclusions?
2) 1. Introduction L24-32. The severest form of muscular dystrophy (MD) termed Duchenne MD (DMD) is a progressive, muscle-wasting disease. First symptoms usually occur at 2 to 3 years of age and most patients become wheelchair dependent when they reach their teens. In addition to accumulating skeletal muscle weakness, respiratory insufficiency and cardiomyopathy develop. The disease is caused by mutations in the protein-coding DMD gene, which leads to the absence of functional dystrophin protein and the disconnection of the intra- cellular muscle fiber cytoskeleton from the extracellular matrix and the basal lamina. Via complex cascading pathways, sarcolemmal damage accumulates, resulting in muscle fiber necrosis and replacement with connective and fatty tissue. Please, support the sentences with references.
3) Introduction. L46-51. A well-balanced diet is paramount to patient wellbeing, yet the progressive nature of DMD disease poses particular challenges. Many patients struggle with constipation and weight gain due to progressive muscle weakness and limited mobility. The prolonged therapeutic use of glucocorticoids further adds to the risk of developing obesity and re-lated endocrine complications. Caloric intake needs to be balanced with physical activity to manage weight throughout the different disease stages, requiring personalized dietary advice and strategies adjusted throughout the patient’s life. I suggest to add some references to support these sentences.
4) L 65-66. This review will focus specifically on the therapeutic potential of amino acid supple- mentation, by evaluating the available preclinical data obtained in the standard disease 66 model, i.e. the mdx mouse. Could you please improve the description of study aim?
5) L248-254. 5. Adverse effects and human clinical trials The side effects of commonly used amino acid supplements have been thoroughly examined in a recent review, concluding that enhanced intake of amino acids is by no means risk-free and may entail side effects, especially at high doses or prolonged use [74]. In general, intravenous administration of amino acid supplements may causes a more sig- nificant proportion of adverse effects when compared to oral or inhaled administration. For DMD, further research is necessary to evaluate the beneficial and possible adverse effects of supplements, and their interactions with therapeutic interventions. Improve this paragraph.
6) 6. Conclusions. Improve the description of conclusions and underline the implications of the study.
Author Response
Dear reviewer,
Thank you for your valued opinion on the submitted manuscript. I have revised the text according to the stylistic comments and requests for added content.
1) Abstract. L16-19. Beneficial effects include improved muscle function and reduced muscle fiber necrosis and reduced inflammatory and oxidative damage to muscle tissues, however mild side effects have also surfaced. In order to identify amino acid formulae that are safe and of true benefit to DMD patients, more explorative, placebo-controlled and long-term clinical trials need to be conducted, documenting therapeutic outcomes. Could you please improve the description of conclusions?
Abstract tekst has been corrected, better explaining the value of pre-clinical studies and the need for follow-up and subsequent human clinical trials.
2) 1. Introduction L24-32. The severest form of muscular dystrophy (MD) termed Duchenne MD (DMD) is a progressive, muscle-wasting disease. First symptoms usually occur at 2 to 3 years of age and most patients become wheelchair dependent when they reach their teens. In addition to accumulating skeletal muscle weakness, respiratory insufficiency and cardiomyopathy develop. The disease is caused by mutations in the protein-coding DMD gene, which leads to the absence of functional dystrophin protein and the disconnection of the intra- cellular muscle fiber cytoskeleton from the extracellular matrix and the basal lamina. Via complex cascading pathways, sarcolemmal damage accumulates, resulting in muscle fiber necrosis and replacement with connective and fatty tissue. Please, support the sentences with references.
Key recent references Szabo et al. and Duan et al. have been added as new references [1] and [2].
3) Introduction. L46-51. A well-balanced diet is paramount to patient wellbeing, yet the progressive nature of DMD disease poses particular challenges. Many patients struggle with constipation and weight gain due to progressive muscle weakness and limited mobility. The prolonged therapeutic use of glucocorticoids further adds to the risk of developing obesity and re-lated endocrine complications. Caloric intake needs to be balanced with physical activity to manage weight throughout the different disease stages, requiring personalized dietary advice and strategies adjusted throughout the patient’s life. I suggest to add some references to support these sentences.
Davis et al., an elegant and thorough review of dietary considerations in DMD, has been added as new reference 6.
4) L 65-66. This review will focus specifically on the therapeutic potential of amino acid supple- mentation, by evaluating the available preclinical data obtained in the standard disease model, i.e. the mdx mouse. Could you please improve the description of study aim?
Study aim has been rephrazed to: “The purpose of this review is to evaluate the therapeutic potential of amino acid supplementation for DMD by analyzing available preclinical data obtained in the standard disease model, i.e. the mdx mouse.”
5) L248-254. 5. Adverse effects and human clinical trials The side effects of commonly used amino acid supplements have been thoroughly examined in a recent review, concluding that enhanced intake of amino acids is by no means risk-free and may entail side effects, especially at high doses or prolonged use [74]. In general, intravenous administration of amino acid supplements may causes a more sig- nificant proportion of adverse effects when compared to oral or inhaled administration. For DMD, further research is necessary to evaluate the beneficial and possible adverse effects of supplements, and their interactions with therapeutic interventions. Improve this paragraph.
The paragraph has been rewritten: “Amino acid supplements are commonly used by healthy individuals as well as patients. However, enhanced intake of amino acids is by no means risk-free and may entail side effects, especially at high doses or after prolonged use. Intravenous administration of amino acid supplements may cause more significant adverse effects when compared to oral or inhaled administration. For DMD, further research is necessary to determine which amino acid supplements are beneficial, and what their interactions are with therapeutic interventions.”
6) 6. Conclusions. Improve the description of conclusions and underline the implications of the study.
The conclusion paragraph has been rewritten: “The desired properties of supportive therapies for DMD include anti-inflammatory [97], anti-oxidant [98] and anti-fibrotic [99] activities. Amino acids and derivatives have been shown to possess combinations of these characteristics, making them plausible candidates as disease-modifying nutraceuticals. Preclinical studies in the mdx mouse disease model have increased our understanding of benefits and associated side effects, and have identified promising candidates. Arg, NAC and Tau are among the most intensively investigated, and their compatibility with current standard of care glucocorticoid and exon-skipping therapies have been investigated to some extent. Only a limited set of supplements have, however, been tried in humans up to now, therefore not enough evidence is currently available to propose amino acid formulae of true benefit to DMD patients. More explorative, placebo-controlled and long-term clinical trials are required.”
Round 2
Reviewer 1 Report
The author addressed all comments appropriately. The work may be accepted.
Reviewer 3 Report
The manuscripts has been improved. No further comments.